# Effect of Sleep Disturbance on Cognitive Function in Elderly Individuals: A Prospective Cohort Study

**DOI:** 10.3390/jpm12071036

**Published:** 2022-06-24

**Authors:** Dong-Kyu Kim, Il Hwan Lee, Byeong Chan Lee, Chang Youl Lee

**Affiliations:** 1Department of Otorhinolaryngology-Head and Neck Surgery, Chuncheon Sacred Heart Hospital, Hallym University College of Medicine, Chuncheon 24253, Korea; doctordk@naver.com (D.-K.K.); leeent@hallym.or.kr (I.H.L.); bchan621@hallym.or.kr (B.C.L.); 2Institute of New Frontier Research, Division of Big Data and Artificial Intelligence, Chuncheon Sacred Heart Hospital, Hallym University College of Medicine, Chuncheon 24253, Korea; 3Division of Pulmonary, Allergy and Critical Care Medicine, Department of Internal Medicine, Chuncheon Sacred Heart Hospital, Hallym University College of Medicine, Chuncheon 24253, Korea

**Keywords:** cognitive function, geriatric patients, sleep disturbance

## Abstract

Many epidemiologic and clinical studies have shown significant links between the degree of sleep disturbance and severity of impairment of selective cognitive functions, including the risk of neurodegenerative diseases. However, the sleep parameters that affect cognitive function in old age are unclear. Therefore, we investigated the association between sleep parameters and cognitive function in older patients. Patients aged above 65 years who complained of sleep-disordered breathing were enrolled consecutively. The Mini-Mental-State Examination tool was used to evaluate cognitive function. Eighty patients (normal cognitive function, *n* = 32 and cognitive impairment, *n* = 42) were included in this study. Multiple linear regression and binary logistic regression analyses were performed to explain the relationship between sleep parameters and cognitive function. We found that the body mass index (BMI) was significantly lower in the cognitive impairment group than in the normal cognitive function group. Additionally, the cognitive impairment group showed significantly decreased sleep efficiency and an increased apnea index compared with normal subjects. Moreover, lower BMI, reduced sleep efficiency, and high frequency of apnea events during sleep were associated with an increased risk of cognitive impairment.

## 1. Introduction

Sleep patterns usually change during the course of normal aging, where a decrease in total sleep duration and efficiency, an increase in sleep fragmentation, difficulty falling asleep, a decrease in rapid-eye-movement (REM) sleep, and predominance of slow-wave sleep are some of the commonly observed sleep-related problems [1,2]. However, some of these sleep disturbances may be due to neurodegenerative processes. Specifically, the sleep–wake cycle is regulated by interactions between areas of the brain related to memory and cognitive function [3,4,5,6].

A previous study showed that various sleep problems occurred in up to 45% of patients with Alzheimer’s disease and other dementias [7]. Other studies showed that patients who have neurodegenerative disease mainly have symptoms of micro-architectural sleep alteration, nocturnal sleep fragmentation, decreased nocturnal sleep duration, and inversion of the sleep–wake cycle [8,9,10]. In particular, several studies have reported that sleep-disordered breathing (SDB), such as obstructive sleep apnea, is more common in patients with cognitive decline than in patients with normal cognitive function [11,12,13,14]. Therefore, sleep disturbance can be as stressful to patients and caregivers as the cognitive impairment itself, and is a major risk factor for early hospitalization [15]. Meanwhile, some studies have suggested that there is a bi-directional relationship between sleep disturbances and cognitive impairment [16,17]. One meta-analysis study showed that sleep disturbances such as insomnia and SDB were risk factors for dementia [18].

However, these studies have the following limitations: most of the patients’ sleep information was obtained from self-report or actigraphy, and the measurements of cognitive function were not constant. Therefore, in aiming to examine the relationship between sleep and cognitive function in old age, we consecutively enrolled older patients with SDB and compared sleep patterns between the groups with normal cognitive function and cognitive impairment using laboratory polysomnography (PSG) and Mini-Mental State Examination (MMSE) results.

## 2. Materials and Methods

### 2.1. Patients

This study was approved by the Institutional Review Board of Hallym Medical University Chuncheon Sacred Hospital (No. 2019-10-003). Patients over 60 years old who complained of SDB were consecutively and prospectively recruited at Chuncheon Sacred Hospital from October 2020 to April 2021. All participants provided written informed consent. We excluded patients who had a history of neurodegenerative disease, had undergone surgeries for sleep disorders, and had serious comorbidities (e.g., cancer, severe depression, severe cardiac or respiratory failure, severe renal or hepatic insufficiency). The patients were divided into two groups according to the results of cognitive function assessments: normal cognitive function group and cognitive impairment group.

### 2.2. Cognitive Outcome Measures

The MMSE is used to assess global cognitive functions [19]. MMSE scores range from 0 to 30, and the higher the score, the better the cognitive function. In this study, we used the Korean version of the Mini-Mental-State Examination (MMSE-KC), which was modified for the socio-cultural and language characteristics of the Korean population [20]. MMSE-KC scores are influenced by age, sex, and education level. Therefore, cognitive impairment is evaluated using normative scores for age, sex, and education level. Scores of 25 or higher were classified as normal results; however, scores below 24 were classified as abnormal results, indicating the possibility of cognitive impairment.

### 2.3. Polysomnography

Standard overnight polysomnography (PSG) was performed in the sleep laboratory of Chuncheon Sacred Hospital, using a computerized polysomnographic device (Nox-A1; Nox Medical, Reykjavik, Iceland). The sleep stage and respiratory events were scored according to the guidelines of the American Academy of Sleep Medicine [21,22]. The following sleep parameters were assessed: total sleep time (TST), sleep efficiency (TST/time in bed × 100), sleep stages (stages N1, N2, N3, and R), oxygen desaturation index (ODI), apnea index (AI), hypopnea index (HI), and apnea–hypopnea index (AHI). ODI was defined as the number of episodes of oxygen desaturation per hour of sleep, with oxygen desaturation defined as a decrease in blood oxygen saturation (SpO_2_) to lower than 3% below baseline. Apnea was defined using an oronasal thermal sensor when the peak signal excursions dropped by ≥90% of the pre-event baseline, lasting for more than 10 s. Hypopnea was scored when the peak signal excursions using nasal pressure dropped by ≥30% of the pre-event baseline for at least 10 s, followed by a ≥3% decrease in oxygen desaturation or accompanied by arousal. AI or HI was defined as the number of apneas or hypopneas per hour of sleep, and AHI was defined as the sum of the number of apneas and hypopneas per hour of sleep. Based on the AHI, the severity of obstructive sleep apnea (OSA) was classified as follows: no OSA, AHI < 5 per hour; mild OSA, 5 ≤ AHI < 15 per hour; moderate OSA, 15 ≤ AHI < 30 per hour; severe OSA, AHI ≥ 30 per hour.

### 2.4. Statistical Analysis

Numerical variables were expressed as means ± standard deviations. We used the normality test to determine whether sample data were drawn from a normally distributed population and then, Student’s *t*-test was used to compare demographic factors and the scores of sleep parameters (TST, sleep efficiency, AI, HI, AHI, ODI, and snoring) between the two groups. Chi-square tests were used to compare categorical variables (sex, medical history, severity of OSA). As a predictive analysis, multiple linear regression and binary logistic regression analyses were performed to determine the relationship between sleep parameters and cognitive function. The area under the receiver operating characteristic curve (AUC ROC) was calculated to assess the diagnostic performance of the variables. The AUC ROC was computed by plotting sensitivity versus 1-specificity to evaluate the ability of each variable’s score to discriminate cognitive impairment. All statistical analyses were conducted using R version 3.5.0 (R Foundation for Statistical Computing, Vienna, Austria). *p* < 0.05 was considered statistically significant.

## 3. Results

### 3.1. Demographic, Cognitive Function, Sleep-Related Problem in the Study Population

Eighty patients (21 [26.2%] men and 59 [73.8%] women) were included in this study. The mean patient age was 73.1 ± 5.63 years (range, 60–85 years). The patients were divided into a group with normal cognitive function (32 patients) or a group with cognitive impairment (48 patients), according to the MMSE score. The mean MMSE score of the cognitive impairment group was 20.58 ± 2.74, and the mean MMSE score of the normal cognitive group was 26.53 ± 1.22. The baseline characteristics are summarized in Table 1. There were no significance differences in demographic variables, comorbidities, Epworth sleepiness scale scores, and sleep apnea severity; however, BMI was significantly lower in the cognitive impairment group (*p* < 0.001, 95% CI: 1.56–4.63).

### 3.2. Polysomnographic Data of Patients with and without Cognitive Impairment

We compared the sleep profiles of the two groups obtained by PSG (Table 2). The cognitive impairment group had lower TST than the normal cognitive function group (240 and 259 min, respectively), but there was no significant difference (*p* = 0.061). Further, the cognitive impairment group showed a significantly decreased sleep efficiency (*p* = 0.047) and increased AI (*p* = 0.038). However, there was no significant difference between the groups with regard to sleep stage architecture, AHI, ODI, and oxygen saturation.

### 3.3. Relationship between Sleep-Related Factor and Cognitive Function

Multiple linear regression analysis was carried out with cognitive function (MMSE score) as the dependent variable, and BMI, sleep efficiency, and AI as the independent variables, after analysis of covariance. BMI and sleep efficiency were positively related to normal cognitive function, whereas high AI was associated with reduced cognitive function (Table 3). In the binary logistic regression analysis, BMI, sleep efficiency, and AI were significantly associated with cognitive function (odds ratio [OR] = 0.746, 95% confidence interval [CI] = 0.598–0.892; OR = 1.965, 95% CI = 1.007–2.360; OR = 2.028, 95% CI = 1.076–3.531, respectively) (Table 4). AI showed the strongest relationship with cognitive function. In the ROC curve analyses, the area under the ROC curve for the prediction of cognitive impairment using BMI, sleep efficiency, and AI was 0.794, with a sensitivity of 66.7% and specificity of 85.7% (Figure 1).

## 4. Discussion

As people age, their sleep patterns naturally change and cognitive function declines. An estimated 25% of the older population experiences sleep problems, and 45% of patients with cognitive impairment have been reported to experience sleep disturbance [7,23]. Sleep disturbances in patients with cognitive impairment are qualitatively similar to those in older people but much more severe [9,24]. Several studies have been conducted from various perspectives to determine the relationship between sleep and cognition. In these studies, patients with cognitive impairment showed increased wakefulness after sleep onset and increased sleep latency, but reduced TST, sleep efficiency, and REM sleep [1,8,25]. To our knowledge, this study is the first to investigate the effect of sleep disturbance on cognitive function in old age using objective parameters. BMI, among demographic factors, and sleep efficiency and AI, among sleep parameters, were significantly associated with cognitive function. Interestingly, BMI and sleep efficiency were significantly lower in patients with cognitive impairment, and as the AI score increased, the tendency to developing cognitive decline also increased.

Previous studies have reported that short sleep duration and reduced sleep quality in patients with cognitive decline are associated with an increase in amyloid-β accumulation, one of the well-known risk factors of Alzheimer’s disease [26,27]. A functional imaging study using positron emission tomography showed a correlation between short sleep duration (or poor sleep quality) and high amyloid-β burden in 70 healthy older subjects [27]. Other studies reported that 40–70% of the patients with Alzheimer’s disease showed an AHI of 5 or higher and were at a higher risk of SDB than healthy older subjects [28,29]. Additionally, other studies showed that high AHI and hypoxemia were related to low global cognitive function [30,31,32]. Consistent with the results of these studies, we found that patients with cognitive impairment experienced significantly more apnea events during sleep, with lower sleep efficiency than patients with normal cognitive function.

Cognitive decline caused by SDB did not have an equal effect on all cognitive domains, and in particular, vigilance, executive functions, and memory domains were related [33]. Increased sleep fragmentation and hypoxia are two mechanisms by which SDB can impair cognition. An animal study showed that hypoxia was associated with an increase in neuronal apoptosis and hippocampal atrophy via oxidative stress and inflammatory pathways [34,35]. Other studies showed that exposure to hypoxia increased the levels of key components of AD, such as amyloid-β, amyloid plaques in the brain, and tau phosphorylation [36,37]. Similarly, in one study with a cognitive healthy APOE-ε4 negative adult, intermittent hypoxia was associated with an increase in the levels of phosphorylated tau, total tau, and amyloid-β in the cerebrospinal fluid [38]. Meanwhile, in neuroimaging studies, patients with obstructive sleep apnea showed decreased white matter integrity and grey matter volume in areas related to memory and execution function compared with healthy control [39,40]; in a large neuroimaging study of more than 800 older adults, SDB showed a strong association with longitudinal white matter changes, suggesting that the SDB increased the risk of cognitive impairment via changes in the white matter caused by vascular disease [41].

Our study had several strengths. First, the patients were recruited consecutively and prospectively. Second, we used laboratory PSG, which is the gold standard for diagnosing obstructive sleep apnea. We were able to evaluate sleep patterns and apnea-hypopnea events during sleep more accurately than previous studies that used actigraphy or home PSG. However, this study also has certain limitations. First, we used the MMSE to evaluate cognitive function. However, it is a relatively simple assessment for cognitive function. To confirm the impact of sleep disturbance on a specific cognitive domain, more detailed and specific cognitive function assessments are required. Second, some demographic factors closely related to cognitive function, such as household income and educational level, were not controlled [42]. Therefore, our findings might have entailed confounding biases. Future studies with larger sample sizes are needed to clarify the relationship between sleep parameters and cognitive function.

## 5. Conclusions

BMI and sleep efficiency positively correlate with cognitive function, but AHI scores are negatively associated with cognitive function in old age. Therefore, clinicians should pay attention to early detection of cognitive impairment when treating older patients with SDB who show low sleep efficiency and a high frequency of apnea events.

## Figures and Tables

**Figure 1 jpm-12-01036-f001:**
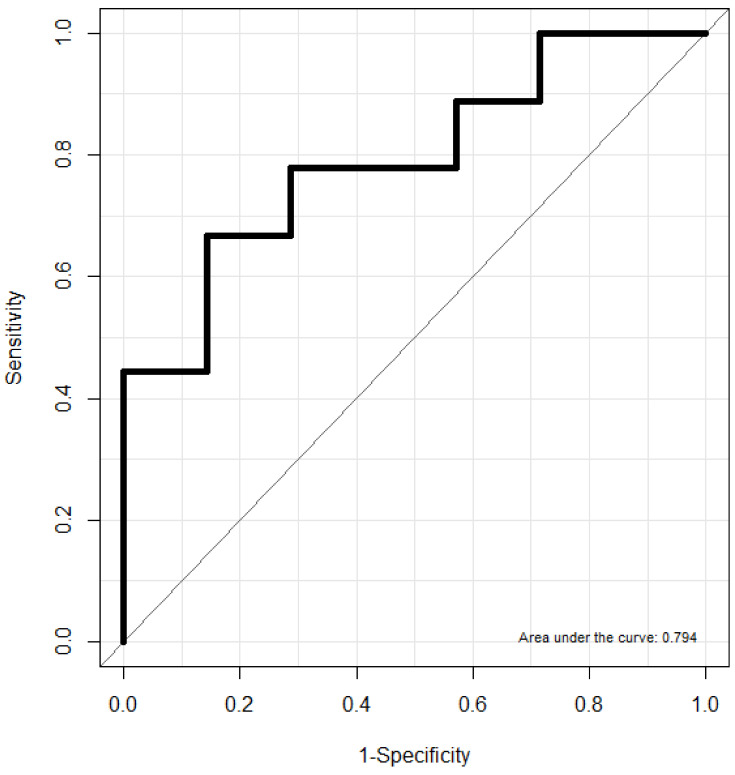
Receiver operating characteristic curve (ROC) for prediction of cognitive impairment using BMI, sleep efficiency and apnea index.

**Table 1 jpm-12-01036-t001:** Characteristics of study population.

	Patients with Normal Cognitive Function (*n* = 32)	Patients with Cognitive Impairment (*n* = 48)	*p* Value
Age	71.03 ± 4.97	74.54 ± 5.65	0.006
Sex (male)	8 (40%)	13 (27.1%)	1.000
BMI (kg/cm^2^)	26.68 ± 3.91	23.59 ± 2.97	<0.001
Neck circumference (cm)	35.72 ± 2.61	34.44 ± 2.72	0.040
Waist circumference (cm)	94.44 ± 10.77	90.06 ± 8.67	0.048
Hip circumference (cm)	96.67 ± 7.57	91.79 ± 6.35	0.003
Hypertension (yes)	18 (56.3%)	25 (52.1%)	0.820
Diabetes (yes)	7 (21.9%)	16 (33.3%)	0.320
Stroke (yes)	0 (0%)	2 (4.2%)	0.514
MMSE-KC score	26.53 ± 1.22	20.58 ± 2.74	<0.001
ESS score	7.53 ± 4.09	6.40 ± 4.62	0.263
Severity of OSA			0.606
Normal	0	0	
Mild OSA	5	4	
Moderate OSA	10	15	
Severe OSA	17	29	

BMI, body mass index; MMSE-KC, Korean version of Mini-Mental State Examination; ESS, Epworth sleepiness scale; OSA, obstructive sleep apnea.

**Table 2 jpm-12-01036-t002:** Comparison of sleep profiles between normal and cognitive impairment groups.

Polysomnographic Data	Patients with Normal Cognitive Function	Patients with Cognitive Impairment	Effect Size	*p* Value
TST (min)	259.76 ± 41.78	241.12 ±43.59	0.436	0.061
Sleep efficiency (%)	70.82 ± 12.74	64.60 ± 13.99	0.462	0.047
Sleep stage				
N1 (%TST)	31.97 ± 19.40	31.80 ± 18.67	0.005	0.970
N2 (%TST)	51.33 ± 21.17	52.53 ± 17.11	−0.064	0.780
N3 (%TST)	2.52 ± 4.73	3.24 ± 5.48	−0.139	0.545
REM (%TST)	14.21 ± 7.05	12.37 ± 8.43	0.233	0.312
AI (event/h)	5.99 ± 12.05	12.71 ± 16.34	−0.456	0.038
HI (event/h)	29.26 ± 15.91	26.86 ± 12.38	0.173	0.452
AHI (event/h)	35.26 ± 22.44	39.58 ± 21.33	−0.198	0.388
Supine AHI (event/h)	42.61 ± 25.42	51.92 ± 27.95	−0.346	0.134
Lateral AHI (event/h)	17.87 ± 22.74	16.46 ± 18.84	0.069	0.766
Mean SaO_2_ (%)	92.40 ± 1.99	93.05 ± 1.41	−0.384	0.120
Lowest SaO_2_ (%)	82.97 ± 5.29	83.53 ± 4.52	−0.116	0.612
ODI (event/h)	16.5 ±12.52	16.83 ± 11.54	−0.027	0.905
Snoring (%)	22.62 ± 18.41	28.13 ± 20.57	−0.278	0.225

TST, total sleep time; REM, rapid-eye-movement; AI, apnea index; HI, hypopnea index; AHI, apnea-hypopnea index; ODI, oxygen desaturation index; SaO_2_, oxygen saturation.

**Table 3 jpm-12-01036-t003:** Result of multiple linear regression analysis of the contribution of other variables on cognitive function.

Variable	Unstandardized Coefficients		Standardized Coefficients	*T*	*p* Value
	B	Standard Error	Beta		
Age	−0.103	0.071	−0.161	−1.446	0.152
BMI	0.322	0.104	0.333	3.106	0.003
Sleep efficiency	0.123	0.068	0.335	2.758	0.014
Apnea index	−0.041	0.102	−0.087	−2.336	0.046

**Table 4 jpm-12-01036-t004:** Determinants of cognitive impairment on binary logistic regression analysis results.

	B	Standard Error	Odds Ratio	95% Confidence Interval
Age	0.075	0.051	1.077	(0.977, 1.195)
BMI	−0.290	0.094	0.749	(0.611, 0.886)
Sleep efficiency	−0.301	0.023	1.645	(1.034, 1.673)
Apnea index	0.270	0.021	1.824	(1.006, 2.935)

## Data Availability

The datasets generated and/or analyzed in the present study are not publicly available but are available from the corresponding author upon reasonable request.

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
