# Peer review of "Effect of Sleep Disturbance on Cognitive Function in Elderly Individuals: A Prospective Cohort Study"

_jpm, 2022, doi:10.3390/jpm12071036_

Round 1
Reviewer 1 Report
The sample were 80 patients aged 60–85 years with sleep-disordered breathing (SDB) without serious comorbidities (e.g., severe depression). The patients were divided into normal (n = 32) or impaired (n = 48) cognitive functioning according to a Mini-Mental-State Examination (MMSE) score above or below 24 (scale 0–30). Results indicated that the cognitive impairment group showed a significantly decreased sleep efficiency and increased apnea index (AI). Also, in logistic regression analysis, sleep efficiency and AI as well as BMI were significantly associated with cognitive function. Overall, the paper is well-written, the theoretical background and motivation are coherent, and the statistical analyses are sound. The methods are a strength of the study (e.g., laboratory polysomnography). The only major drawback of the study is the cross-sectional design. In my view, the questions that are raised and answered by this paper are nevertheless relevant to the journal's readership. I only have a few issues:
1. Inconsistency: Under 2.1. patients (page 2, line 59) participants are described as being over 65 years of age, but the observed range is 60–85.
2. „Almost significant“ (page 3, line 125) does not exist. A mean difference is either statistically significant or not.
3. Table 2: Please report effect sizes for significant differences (e.g., Cohen’s d)
4. Please consider using age as a control variable in your regression analyses
Author Response
1. Inconsistency: Under 2.1. patients (page 2, line 59) participants are described as being over 65 years of age, but the observed range is 60–85.
Answer: Thank you for your comment. This is our mistake and we modified this as follows: “Patients over 60 years old who complained of SDB were consecutively ~”.
2. Almost significant (page 3, line 125) does not exist. A mean difference is either statistically significant or not.
Answer: Thank you for your comment. We modified this as follows: “~ but there was no significant difference (P = 0.061)”.
3. Table 2: Please report effect sizes for significant differences (e.g., Cohen’s d)
Answer: As you commented, we added the effect size in Table 2.
4. Please consider using age as a control variable in your regression analyses
Answer: As you recommended, we reanalyzed Table 3 and Table 4 after including the age variable.

Reviewer 2 Report
1. The normality of data should be checked and then the analysis method is selected.
2. For categorical data, percentages must be presented as well as counts.
3. All p-values better presented in 3 digits regarding the APA protocol.
4. Why was BMI bold in table 3?
Author Response
1. The normality of data should be checked and then the analysis method is selected.
Answer: Thank you for your comment. In this study, we used the normality test to determine whether sample data has been drawn from a normally distributed population but we missed this description in the section on methodology. We added this description to the methodology.
2. For categorical data, percentages must be presented as well as counts.
Answer: As you commented, we added percentages in Table 2.
3. All p-values better presented in 3 digits regarding the APA protocol.
Answer: As you commented, we modified all p-values expressions as 3 digits.
4. Why was BMI bold in table 3?
Answer: This is our mistake. So, we modified this.
